# Association of *KRAS* Mutation and Gene Pathways in Colorectal Carcinoma: A Transcriptome- and Methylome-Wide Study and Potential Implications for Therapy

**DOI:** 10.3390/ijms25158094

**Published:** 2024-07-25

**Authors:** Farzana Jasmine, Armando Almazan, Yuliia Khamkevych, Marc Bissonnette, Habibul Ahsan, Muhammad G. Kibriya

**Affiliations:** 1Institute for Population and Precision Health, Biological Sciences Division, The University of Chicago, Chicago, IL 60637, USA; armando.almazan@bsd.uchicago.edu (A.A.); yuliiak@uchicago.edu (Y.K.); hahsan@bsd.uchicago.edu (H.A.); kibriya@uchicago.edu (M.G.K.); 2Department of Medicine, The University of Chicago, Chicago, IL 60637, USA; mbissonn@bsd.uchicago.edu; 3Department of Public Health Sciences, Biological Sciences Division, The University of Chicago, Chicago, IL 60637, USA

**Keywords:** colorectal carcinoma, *KRAS* mutation, MSI, *CDKN2A*, Villosol, proteasome inhibitor, *VEGF*, *EGFR*

## Abstract

Kirsten Rat Sarcoma (*KRAS*) is the most commonly mutated oncogene in colorectal carcinoma (CRC). We have previously reported the interactions between microsatellite instability (MSI), DNA promoter methylation, and gene expression. In this study, we looked for associations between *KRAS* mutation, gene expression, and methylation that may help with precision medicine. Genome-wide gene expression and DNA methylation were done in paired CRC tumor and surrounding healthy tissues. The results suggested that (a) the magnitude of dysregulation of many major gene pathways in CRC was significantly greater in patients with the *KRAS* mutation, (b) the up- and down-regulation of these dysregulated gene pathways could be correlated with the corresponding hypo- and hyper-methylation, and (c) the up-regulation of *CDKN2A* was more pronounced in tumors with the *KRAS* mutation. A recent cell line study showed that there were higher *CDKN2A* levels in 5-FU-resistant CRC cells and that these could be down-regulated by Villosol. Our findings suggest the possibility of a better response to anti-*CDKN2A* therapy with Villosol in *KRAS*-mutant CRC. Also, the more marked up-regulation of genes in the proteasome pathway in CRC tissue, especially with the *KRAS* mutation and MSI, may suggest a potential role of a proteasome inhibitor (bortezomib, carfilzomib, or ixazomib) in selected CRC patients if necessary.

## 1. Introduction

Colorectal cancer (CRC) is the third most commonly diagnosed cancer in the United States and it is the second most common cause of cancer death [1].

Its incidence is increasing in developing countries. An increased incidence rate is also being observed in younger patients under 45 years of age. During the past few years, the mortality rate has been falling in those countries where early screening and treatments have been implemented. This positive outcome reflects earlier detection, advances in imaging, improvements in surgical techniques, and advances in chemotherapy and radiation [2].

Extensive studies have been carried out at the genetic and epigenetic level to explore the molecular mechanism of cancer development and cancer characteristics for targeted cancer therapy. Studies have shown that several cellular signaling pathways are dysregulated in CRC and results vary among different malignant phenotypes. Therefore, analyzing the signaling pathways involved in CRC is necessary to elucidate the underlying mechanism of CRC progression and pharmacotherapy.

Kirsten Rat Sarcoma (*KRAS*) is a known oncogene and the most commonly mutated oncogene in all cancers. When activated, wild-type *KRAS* binds GTP, which results in a conformational change that allows the protein to bind and activate more than 20 known downstream effectors, including Raf, Braf, mTOR, MEK1 and 2, ERK, AKT, and PIK3CA. These downstream effectors have many different effects, including apoptosis suppression, the promotion of cell growth, cell transformation, angiogenesis, migration, and differentiation [3,4,5]. *KRAS* mutations in CRC include G12V, G12D, G12C, and G13D [6,7,8]. Targeting these different variants has been discussed elaborately in recent papers [6,8]. The *KRAS* oncogene is mutated in approximately 39.8% of CRC patients [7]. Another study found that *KRAS* and *NRAS* mutations were found in 36.7% and 2.9% of 210 studied patients, respectively [9]. In our study, we found 28.1% of patients had a *KRAS* mutation [10,11].

In recent years, targeted therapies have been used against vascular endothelial growth factor (*VEGF*) and epidermal growth factor receptor (*EFGR*), which showed improved survival in metastatic CRC. Patients with mutated *KRAS* CRC are unlikely to benefit from anti-EGFR therapy. Furthermore, 40–60% of patients with wild-type KRAS tumors also do not respond to such therapy [12]. The factors affecting the results are the differences in the molecular biology of CRC and the lack of biologic predictive markers to target the appropriate patient populations for these agents [13].

The following different molecular processes are involved in CRC: (a) chromosomal instability or copy number changes [14,15,16], (b) microsatellite instability (MSI) [10,11,17,18], (c) DNA promoter methylation [10,19,20,21], (d) somatic mutation [22,23,24,25,26,27], and (e) change in telomere length [28,29,30]. Therefore, it is important to understand their interactions, not only for better understanding of the pathology, but also for generating a hypothesis for precision medicine based on the molecular findings from clinical samples. We have previously reported the interaction of MSI, gene expression, and methylation in CRC [10,11]. In this study, we asked (a) if the *KRAS* mutation is associated with the differential expression of major gene pathways, (b) if those expression changes are also associated with DNA promoter methylation, and (c) if this molecular profiling can help our understanding of the pathogenesis and provide a molecular basis for selecting patients for any targeted therapy in future.

## 2. Results

A comparison of *KRAS* wild-type CRC and *KRAS*-mutant CRC is presented in Appendix A, and we did not find differences in patient characteristics and MSI status in our patients.

### 2.1. Transcriptome-Wide Analyses at Gene Level

The transcriptome-wide analysis of the gene expression data from tumor–normal paired tissues from the same patient revealed that a total of 531 genes were differentially expressed (down-regulated = 378 and up-regulated = 153) by at least a twofold change (FC) in either direction (up-regulated or down-regulated) with the FDR ≤ 0.05 level in the CRC tumor tissue compared to the corresponding apparently unaffected “normal” tissue (see Appendix A). The enrichment of these significantly differentially expressed genes is shown in Appendix A.

In the next step, we performed a similar analysis in patients with *KRAS* wild-type tumors and separately in patients with the *KRAS* mutation. The overlap of the differentially expressed genes (with similar FC and FDR criteria) is shown in the Venn diagram (see Figure 1), which produced lists of genes that are (a) differentially expressed irrespective of the *KRAS* mutation status (n = 408, see Appendix A), (b) differentially expressed only in tumors with the *KRAS* mutation (n = 245, see Appendix A), and (c) differentially expressed only in tumors with the *KRAS* wild-type (n = 75, see Appendix A). The enrichment analysis of these lists of genes is shown in Appendix A, respectively.

These enrichment analyses of the differentially expressed (FC 2 and FDR 0.05) genes suggested that the genes involved in certain pathways were more frequently seen to be differentially expressed in CRC tissue than would occur by chance. For example, the genes involved in “Th1 and Th2 cell differentiation”, “Th17 cell differentiation”, “p53 signaling pathway”, and “intestinal immune network for IgA production” were differentially expressed in CRC tissue compared to the corresponding normal tissue if the tumor had the *KRAS* mutation.

For *KRAS*, we had two probes on the microarray. Probe ILMN_1728071 targets all known isoforms (probe type A) and probe ILMN_1652104 targets only one isoform of *KRAS* (probe type I). This isoform-specific probe was designed against *KRAS* isoform-A only (NM_033360.2). The differential expression (tumor vs. normal) of both of these probes are presented in Appendix A. For the type-A probe (targeting all known isoforms, and hence representing expression of *KRAS* in general), there was down-regulation of *KRAS* in CRC tissue compared to normal tissue (*p* = 0.0002) irrespective of the *KRAS* mutation status or tumor staging and the magnitude of differential expression was not different based on the KRAS mutation status (interaction *p* = 0.82) or tumor stage (interaction *p* = 0.33). However, for the isoform-specific probe (targeting only isoform A), there was no differential expression (*p* = 0.54).

### 2.2. Association of KRAS Mutation and Genes Related to Tyrosine Kinase Inhibitor Targets and Other Angiogenic Targets

We asked if the magnitude of the differential expression of a given gene/gene probes (the array had multiple gene probes for some genes) was different in the absence or presence of the *KRAS* mutation. The analysis suggested that the cyclin-dependent kinase inhibitor 2A (*CDKN2A,* also known as *p16*) was up-regulated in CRC tissue compared to in normal tissue in general (see Figure 2), but the up-regulation was more marked in tumors with the *KRAS* mutation than in tumors with the *KRAS* wild-type as follows: for the *CDKN2A* probe ILMN_1717714 (probe type-A targeting all known isoforms), FC = 1.42 (95% CI 1.26–1.59) vs. FC = 1.17 (95% CI 1.08–1.26) (interaction *p* = 0.001); and for the *CDKN2A* probe ILMN_1744295 (probe type I, targeting longest isoform variant 4), FC = 1.24 (95% CI 1.13–1.36) vs. FC = 1.09 (95% CI 1.03–1.16) (interaction *p* = 0.04), respectively. The differential expression of another isoform-specific probe for isoform variant 3 (probe ILMN_1757225, designed against NM_058197.3 for NP_478104.2) was not significant (see Figure 2I–L). So, our data suggest that the overall up-regulation of *CDKN2A* was probably reflecting the up-regulation of the longest variant of *CDKN2A* (isoform variant 4). The up-regulation of *CDKN2A* in CRC tissue and the interaction with the *KRAS* mutation status (meaning more up-regulation in the presence of mutation) remained significant even when the tumor stage was taken into account. It may be noted that for all three probes, the “person-to-person” variation (shown in pie charts as “case ID”) contributes a major fraction of the total “source of variation” in the expression data. This emphasizes the importance of examining the tumor and corresponding normal tissue from same patient in such study. This finding of up-regulation of *CDKN2A* is important from a targeted therapy point of view in a sense that *CDKN2A* inhibition using Villosol may be a potentially good strategy especially for tumors with *KRAS* mutation. In fact, a recent study using cell line and mice, demonstrated the beneficial effect of Villosol in mice model [31]. In connection with *CDKN2A*, we also looked for the differential expression of *TP53* (see Appendix A). We found that *TP53* was down-regulated in CRC tissue irrespective of the *KRAS* mutation status or tumor stage.

From an anti-*VEGF* targeted therapy point of view, it may be noted that *VEGFA* was significantly up-regulated in CRC compared to corresponding normal tissue (see Figure 3 for two different probes on the array) both for tumors with the *KRAS* wild-type and tumors with the *KRAS* mutation and the magnitude of up-regulation was not different (interaction *p*-value > 0.05 for both the probes). This suggests potential beneficial effect of anti-*VEGFA* (most commonly used anti-*VEGF* targeted therapy) in CRC and this is in line with the clinical practice as well. *VEGFB* was not significantly changed in CRC with *KRAS* mutation, but was slightly over-expressed in CRC tissue with the *KRAS* wild-type (FC = 1.06 (95% CI 1.02–1.10) suggesting the potential response only in the *KRAS* wild-type.

Regarding the potential of anti-*EGFR* therapy (see Figure 3), we did not find evidence of significant dysregulation of *EGFR* gene in CRC tissue in general or depending on the *KRAS* mutation status that could suggest a beneficial effect of anti-*EGFR* therapy at least in this population.

### 2.3. Transcriptome-Wide Analyses at Gene Pathway Level

In the next step, using Gene set ANOVA, instead of a single gene level comparison, we asked if a set of genes (sharing similar biological pathways, e.g., KEGG pathway) on average was differentially expressed in CRC tissue compared to corresponding normal tissue from same patient. We found that forty-six gene pathways were dysregulated (up-regulated = 11 and down-regulated = 35) by at least 10% in either direction with FDR 0.05 (see Table 1). Examples from among the up-regulated pathways are “DNA replication”, “mismatch repair”, “nucleotide excision repair”, “proteasome” and examples from down-regulated pathways include metabolic pathways like “Nitrogen metabolism”, “Sulfur metabolism”, “Retinol metabolism”, “Fatty acid degradation” and others like “Antigen processing and presentation”. It may be noted that 18 of these pathways were also picked up by the enrichment analysis.

By including an interaction term—“tissue (1 = CRC, 0 = normal) × *KRAS* mutation status (1 = mutation, 0 = wild-type)” in the ANOVA model(s), we also asked if the magnitude of the differential expression of a given pathway was significantly different in the presence or absence of *KRAS* mutation in the tumor. We found that for 26 KEGG pathways (up-regulated = 12, down-regulated = 14; see Table 2), the magnitude of differential expression (CRC tissue vs. normal tissue) was significantly greater in the presence of *KRAS* mutation (ANOVA interaction *p* < 0.05, see Table 2).

We also found that for 30 pathways, the magnitude of differential expression was not statistically different (interaction *p* > 0.05; see Appendix A). It may be noted that all the pathways listed in Table 2 and Appendix A were dysregulated in CRC tissue compared to normal tissue regardless of whether the tumor had *KRAS* mutation or was of the wild-type, but for the pathways in Table 2, the magnitude of differential expression was significantly greater in patients with the *KRAS* mutation than in the *KRAS* wild-type indicating the association between the *KRAS* mutation status and gene expression pathways (see the “interaction *p*” column of Table 2). Interestingly, when we looked for the pathways with significant interaction (pathways in Table 2) among the total list of pathways dysregulated in CRC in general (presented in Table 1), most of the top-ranking pathways in the combined list, in fact, were the pathways with significant interaction with *KRAS* mutation (see the pathways highlighted in the Table 1). More specifically, among the top 10 differentially expressed pathways (by FC), the magnitude of up-regulation of 8 of them (80% of pathways) were significantly greater in tumors with the *KRAS* mutation. In other words, our data suggested that the *KRAS* mutation is associated with the greater magnitude of the differential expression of the major pathways that are altered the most (especially the up-regulated pathways) in CRC and perhaps indicate more pronounced genomic dysregulation. This may explain, to some extent, why patients with CRC showing the *KRAS* mutation do not respond well to conventional treatment compared to those with the *KRAS* wild-type. Such up-regulated pathways include, for example, (a) DNA replication (see Figure 4) and proteasome (see Figure 4)—indicating more proliferation, and (b) mismatch repair (see Figure 4) and nucleotide excision repair—indicating the tissue response to DNA damage.

### 2.4. Association of KRAS Mutation and Genes Related to Immune Checkpoint Inhibitors (ICIs)

In our dataset, the inflamed T cell genes were down-regulated in CRC tissue in general, suggesting that ICIs may not have a satisfactory response. However, the magnitude of down-regulation was slightly more pronounced in tumors with the *KRAS* mutation (FC = −1.34 (95% CI −1.41 to −1.28)) compared to tumors with the *KRAS* wild-type (FC = −1.27 (95% CI −1.31 to −1.23)) (ANOVA interaction *p* = 0.003, see Figure 5), suggesting a lower possibility, if any, of a therapeutic response to ICIs. There was no difference in the magnitude of the differential expression of the genes related to platinum drug resistance.

### 2.5. Effect of Interaction of KRAS Mutation and MSI Status on Differential Expression of Gene Pathways

Considering the fact that both *KRAS* mutation and MSI status are clinically important molecular markers in CRC, we asked how the different combinations of these two molecular markers associate or alter the differential expression of the gene pathways. We divided the CRC patients into four categories (see Table 3) and compared the expression of the different gene pathways of each group of tissues to their corresponding normal tissues. The overall FC (95% CI) of the pathways in each group are presented in Table 3. For example, the presence of the *KRAS* mutation or MSI increases the expression of genes in proteasome pathway. However, when a tumor has both the *KRAS* mutation and MSI, the magnitude of up-regulation was the greatest.

### 2.6. Differential DNA Promoter Methylation of Gene Pathways

In the next step, we asked if the dysregulated gene expression pathways in CRC were also differentially methylated and if the magnitude of differential methylation was also associated with the *KRAS* mutation status. For differential methylation, we used delta beta (beta value of tumor tissue—beta value of corresponding normal tissue). So, a positive delta beta means hyper-methylation and a negative delta beta means hypo-methylation in tumor tissue. We used the similar Gene set ANOVA and the interaction term tissue (1 = CRC and 0 = normal) × *KRAS* mutation status (1 = mutation, 0 = wild-type) as we used in gene expression analysis.

Table 4 presents the Gene set ANOVA analysis for the methylation data of the gene pathways, which were more pronouncedly up-regulated in CRC tissue compared to corresponding normal tissue in the presence of the *KRAS* mutation. It was interesting to observe that, except the last two, all other pathways were less methylated (indicated by negative delta beta) for CRC tissue and the magnitude of hypo-methylation (although very minimum) was significantly greater in tumors with the *KRAS* mutation (indicated by the ANOVA interaction *p*-value).

Table 5 presents the Gene set ANOVA analysis for the methylation data of the gene pathways, which were more pronouncedly down-regulated in CRC tissue compared to corresponding normal tissue in the presence of the *KRAS* mutation. It was interesting to observe that, except for the last two, all other pathways were more methylated (indicated by positive delta beta) for CRC tissue and the magnitude of hyper-methylation (although very minimum) was significantly greater in tumors with the *KRAS* mutation (indicated by the ANOVA interaction *p*-value).

In summary, our methylation and gene expression data suggested the associations of the *KRAS* mutation with both methylation and gene expression in two different ways as follows: (a) for one group of gene pathways, the *KRAS* mutation may be associated with DNA promoter hypo-methylation and the over-expression of genes in those pathways; and (b) for another group of gene pathways, the *KRAS* mutation may be associated with DNA promoter hyper-methylation and the down-regulation of genes in those pathways. The experimental design did not allow us to comment on causality, but these associations logically explain the biological changes in the CRC tissue.

## 3. Discussion

In CRC, the *KRAS* mutation is one of the most frequently encountered somatic mutations. This study addressed the association of the *KRAS* mutation status and the transcriptomic profile in CRC. Our results suggested that a large proportion of the most dysregulated gene pathways in CRC in general are in fact influenced by the *KRAS* mutation status. More precisely, the magnitude of dysregulation (both up- and down-regulation of CRC compared to normal tissue) of the major gene pathways is significantly greater in tumors with the *KRAS* mutation. This may reflect more severe pathology at molecular level to be associated with the *KRAS* somatic mutation.

The molecular data suggested a few points regarding potential targeted therapy, as follows: (a) the more pronounced up-regulation of *CDKN2A* in tumors with the *KRAS* mutation may suggest the possibility of a better response to anti-*CDKN2A* therapy with Villosol in *KRAS*-mutant CRC; (b) the similar magnitude of *VEGFA* up-regulation in tumors with the *KRAS* wild-type or mutation supports the clinical guideline for the use of anti-*VEGF* therapy in both cases; (c) our data from the current study could not find a molecular basis for the potential benefit of anti-*EGFR* therapy in this population; and (d) the up-regulation of genes in the proteasome pathway in the CRC tissue compared to normal tissue may suggest a potential role of a proteasome inhibitor (bortezomib, carfilzomib, or ixazomib) in CRC patients if necessary—particularly in tumors with the *KRAS* mutation or with a combination of the *KRAS* mutation and MSI. These medications are used in multiple myeloma, but, to our knowledge, have not yet been tried in CRC. Potential toxicity is a concern; however, based on our molecular data, in refractory cases, this possibility may be considered for future study, especially in a group of patients with a combination of the *KRAS* mutation and MSI, who may benefit the most. If successful, this would be an application of molecular data in precision medicine.

*CDKN2A* (p16) promoter methylation and the loss of *CDKN2A* expression has been reported in CRC [32]. A study reported that *CDKN2A* hyper-methylation was found in 100% of colon cancer cell lines and 55% of colon cancers, but not in colonic epithelium [33]. The study also reported the association of the *KRAS* mutation and the hyper-methylation of *CDKN2A* [33]. However, one study reports no correlation between the *KRAS* mutation and *CDKN2A* (p16) expression [34]. Another study compared *KRAS* wild-type and *KRAS*-mutant CRC tissue to identify genes that are differentially expressed between these two groups. They identified 30 genes; however, *CDKN2A* was not one of them [35].

Patrinia villosa (PV) is a drug used in traditional Chinese medicine for the treatment of 5-FU-resistant CRC. Using the cell line, Chen et al. [31] showed that the protein expression of *CDKN2A* was higher in the 5-FU-resistant cell line than in the CRC cells. They also showed that high *CDKN2A* was associated with a lower sensitivity of cells to 5-FU. Villosol showed strong affinity for *CDKN2A*. In the cell line experiment, they also documented that (a) by lowering *CDKN2A* expression they could see an increase in the effectiveness of 5-FU in the 5-FU-resistant cell line and (b) increasing the *CDKN2A* expression decreased the effectiveness of 5-FU in CRC cells. In their experiment, Villosol significantly inhibited *CDKN2A* activity in the 5-FU-resistant cell line. They also showed that the inhibition of *CDKN2A* activated the *TP53* [31]. In summary, their cell line study showed that the overexpression of *CDKN2A* activates the PI3K/Akt pathway and induces resistance to 5-FU. Villosol effectively reverses 5-FU resistance by affecting the *CDKN2A-TP53-PI3K*/Akt axis [31]. There is evidence that Villosol may act as a *TP53* expression enhancer, apoptosis agonist, and antineoplastic agent [36]. In our current study, we show that in actual clinical samples from CRC patients, human CRC tumor tissue has significant up-regulation of *CDKN2A* compared to corresponding unaffected colon tissue independent of histological staging, and this up-regulation is more marked in tumors with the KRAS mutation. We also found significant down-regulation of *TP53* in tumor tissue.

*KRAS* normally functions in signal transduction cascades initiated by the binding of the *EGFR*. In other word, *KRAS* is downstream in the cascade. Targeted therapy in CRC with the *KRAS* mutation rarely respond to anti-*EGFR* monoclonal antibodies [37]. Even in the *KRAS* wild-type, a significant proportion of patients fail to respond to anti-*EGFR* therapy. Our data provide one of the possible explanations for this—anti-*EGFR* therapy is supposed to show good therapeutic effect if the *EGFR* is up-regulated in tumor tissue, but we did not find any up-regulation of *EGFR* in CRC irrespective of the *KRAS* mutation status; rather, we found that there was non-significant down-regulation. Other possible explanations include the fact that in many cases of *KRAS* wild-type CRC, there is a small percentage of *KRAS* mutant subpopulations that remain undetected. There are cases of relapse in advanced CRC patients treated with *EGFR*-targeted monoclonal antibody therapy and this involves the outgrowth of previously undetected *KRAS*-mutant tumor cell populations. *EGFR*-targeted therapies that treat predominantly *KRAS* wild-type CRC can create an environment for the outgrowth of *KRAS*-mutant subpopulations, leading to acquired resistance to treatment and relapse [37].

In a preclinical CRC cell line model, there were differences in response to anti-EGFR treatments (both cetuximab and panitumumab) in comparisons between *KRAS* wild-type, *KRAS* G12V, and *KRAS* G13D. In this in-vitro experiment *KRAS* wild-type was more sensitive than *KRAS* G13D. *KRAS* G13D showed more sensitivity than *KRAS* G12V, which showed resistance [38].

In a study of 210 CRC patients, investigators found that the *KRAS* mutation was associated with female gender, left localization, classical adenocarcinoma, vascular invasion, and the presence of positive lymph nodes and advanced disease stage in MSS cases [9]. They also found that *KRAS*-mutated cases had a higher incidence of metastatic disease at diagnosis [9]. Another study involving 277 CRC patients with a follow-up for 5 years looked for a correlation with *KRAS* mutation in CRC with clinical characteristics [7]. They found the *KRAS* mutation in 39.8% of patients and that the *KRAS* mutation showed a correlation with the expression of the *EGFR* gene, primary tumor site, and multiple metastases. The average survival time of CRC patients carrying wild-type *KRAS* and mutant-type *KRAS* was 49.9 months and 50.7 months, respectively [7].

In one cell line study of CRC, an interaction was found between the *KRAS* mutation and *Bmp4*, which plays an important role in the embryonic development of multiple organs, including the nervous system, musculature, skeleton, skin, hair, teeth, kidney, lung, and intestinal tract. In *KRAS*-mutant CRC, *Bmp4* expression is down-regulated and this is mediated through the ERK signaling pathway [39]. Apoptotic protease activating factor 1 (*APAF1*) is important for the mitochondrial apoptotic pathway and has been shown to be down-regulated in many CRC cases [40]. Our data also showed down-regulation in CRC irrespective of the *KRAS* mutation status.

Hong Yan et al. showed that only male CRC patients with the *KRAS* mutation had several altered metabolic pathways that suppress ferroptosis (a non-apoptotic mode of cell death), including glutathione biosynthesis, transsulfuration activity, and methionine metabolism [41]. They also looked at the gene expression data from an additional CRC patient cohort (Gene Expression Omnibus (GEO)) and observed higher expression of *GPX4, FTH1,* and *FTL*, and lower *ACSL4* expression, which is associated with poorer 5-year overall survival only in male patients with *KRAS*-mutant tumors. The data also suggested that male *KRAS*-mutant patients had increased iron dependence [41].

There is emerging evidence that triggering ferroptosis may have an important role in eradicating aggressive malignancies that are resistant to traditional therapies. *KRAS*-mutant CRC cells (HCT116) were treated with cetuximab and β-elemene, a bioactive compound isolated from the Chinese herb Curcumae Rhizoma [42]. Ferroptosis and epithelial-mesenchymal transformation (EMT) were detected in vitro and in vivo. In vitro, this combination was shown to induce iron-dependent reactive oxygen species (ROS) accumulation, glutathione (GSH) depletion, lipid peroxidation, up-regulation of HO-1 and transferrin, and down-regulation of negative regulatory proteins for ferroptosis (*GPX4, SLC7A11, FTH1*, glutaminase, and *SLC40A1*) in *KRAS*-mutant CRC cells. This treatment synergistically showed inhibited cell migration and decreased the expression of mesenchymal markers (Vimentin, N-cadherin, Slug, Snail, and *MMP-9*), but promoted the expression of the epithelial marker E-cadherin. In vivo, co-treatment with β-elemene and cetuximab inhibited *KRAS*-mutant tumor growth and lymph node metastases [42].

Hassan et al. studied the relationship of the *KRAS* mutation with ERK, ATK, and the P65 marker expression in CRC [43]. Individuals with high expression of ERK and ATK demonstrated a higher prevalence of positive *KRAS* mutations (53.12% and 62.96%, respectively) but not to the level of significance, indicating that ERK and AKT expression might not be a strong predictor of the presence of the *KRAS* mutation [43], whereas individuals with high expression of the P65 marker exhibited a significantly higher likelihood of having positive *KRAS* mutations (75.0%) in comparison to those with low P65 expression (51.85%), with a statistically significant *p*-value of 0.034, indicating that P65 may play a role in *KRAS*-driven oncogenic processes [43].

The immunohistochemistry of CRC tissue with the *KRAS* mutation showed expression of CD44 and CD166, cancer stem cell markers that are responsible for tumor cell transformation, growth, and proliferation with a higher risk of lymph node involvement by the tumor and the development of liver and lung metastasis [44].

A study has suggested a prognostic signature based on 36 genes involved in CRC with the *KRAS* mutation with good predictive efficiency. The genes are *APOL3, AUTS2, CD6, COG2, CUL1, CYP1B1, DBI, DHX29, FDFT1, GATA6, IFI27, LHPP, LONRF3, MEST, MMRN2, MRPS35, NLE1, NUP107, PLA2R1, POP4, PPP1R8, PSMG2, RHOBTB1, RHOF, S1PR2, SHC1, SLC2A5, SLCO2A1, SNRNP35, SSBP2, SYNGR1, TMEM159, TRIT1, VPS28, YTHDF2,* and *ZNF552* [45].

Xiaorui Fu et al. obtained somatic mutation data and clinical information on CRC samples from the TCGA database (n = 399) and ICGC database [46]. They compared the immune cell infiltration, tumor mutational burden, HLA gene expression, and checkpoint-related genes between the *KRAS*-mutated and wild-type samples. They showed 12 HLA genes had significantly lower expression levels in the *KRAS*-mutated group compared to the wild-type group [46]. The checkpoint-related genes (*BTLA, CD80, CD86, CTLA4, IDO1, PDCD1LG2,* and *TIGIT*) had decreased expression in the *KRAS*-mutated group. With the exception of the *mTOR* and *ERBB* signaling pathways, most of the RAS-related pathways positively correlated with the immune signature. The RAS (r = 0.61) and *FOXO* (r = 0.5) signaling pathways had strong positive correlations with neutrophils. The RAS (r = 0.53) and PI3K-ATK signaling pathways (r = 0.56) exhibited a positive correlation with macrophages [46].

In one retrospective study, the prognostic value of *MACC1* expression and its relation to *KRAS* G12 or G13 mutations for CRC metastasis was analyzed. They showed that only high *MACC1* expression (HR: 6.09, 95% CI: 2.50–14.85, *p* < 0.001) and the *KRAS* G13 mutation (HR: 5.19, 95% CI: 1.06–25.45, *p* = 0.042) were independent prognostic markers for shorter metastasis-free survival. Accordingly, Cox regression analysis revealed that patients with high *MACC1* expression and *KRAS* G13 mutation exhibited the worst prognosis (HR: 14.48, 95% CI: 3.37–62.18, *p* < 0.001) [47].

A study showed that the mutation frequency of *APC* and *PIK3CA* was significantly increased in the *KRAS*-mutant group, while the mutation frequency of *TP53* and *ZFHX4* was significantly increased in the *KRAS* wild-type group [48]. Several immune-related pathways were significantly down-regulated in *KRAS*-mutant CRC compared to *KRAS* wild-type CRC, namely, T1 and T2 cell differentiation, T cell receptor signaling, and nuclear factor kappa-B (*NF-κB*) signaling pathways. Four pathways were significantly up-regulated in the *KRAS*-mutant patients compared to the *KRAS* wild-type CRC patients, namely, the biosynthesis of amino acids, carbon metabolism, oxidative phosphorylation, and ribosome [48].

The *SMARCA4* gene provides instructions for making a protein called BRG1, which is a subunit of the SWI/SNF protein complexes. The SWI/SNF complexes are involved in many processes like repairing damaged DNA, copying DNA, and controlling cellular growth and differentiation. Thus, the BRG1 protein, a product of the *SMARCA4* gene, acts as a tumor suppressor along with other SWI/SNF subunits [49]. *SMARCA4* expression was found to be greater in CRC compared to normal tissue, especially in the *KRAS*-mutant group. The potential role of *SMARCA4* in different cancers was analyzed from available databases. Peng et al. showed that *SMARCA4* is correlated with the prognosis of patients with cancer and immune infiltration across diverse cancers. *SMARCA4* gene expression is associated with MMR, MSI, TMB, and DNA methylation in multiple cancers. *SMARCA4* gene expression was strongly associated with gene expression related to immunity in various cancers. *SMARCA4* may play a key role as a prognostic biomarker [50].

Investigators have tried to explore the methylation alterations in CRC. The methylation of two genes—*ZNF132* and *ESR1*—was recognized as a promising diagnostic biomarker for CRC. Moreover, the study suggested significantly higher diagnostic abilities of *ZNF132* and *ESR1* in the *KRAS*-mutant group than in the *KRAS* wild-type group [51].

In one study of CRC patients, the *MINT2, p16INK4a*, and *p14ARF* genes were more methylated in the *KRAS-* and *BRAF*-mutant group compared to the wild-type group. However, the methylation of *MGMT* was only associated with the *KRAS*-mutant group [52].

Based on the top 10% of probes with the highest DNA methylation in CRC, investigators performed unsupervised clustering using a recursively partitioned mixture model (RPMM). The cluster 1 subgroup was enriched for CIMP-positive CRC (based on a five- marker panel comprising *CACNA1G, IGF2, NEUROG1, RUNX3,* and *SOCS1*). All of the tumors with a *BRAF* mutation belong to this subgroup, and nearly half of the tumors in this subgroup that do not harbor *BRAF* mutations carry mutant *KRAS*. The researchers found that although *KRAS*-mutant tumors are represented across the four classes, they are more common in the cluster 2 subgroup compared with the other clusters and are associated with the CIMP-low group [53].

*KRAS* G12C direct inhibitors have recently been used [6]. These are the FDA-approved drug AMG 510, known as sotorasib, developed by Amgen [54], and MRTX849, known as adagrasib, developed by Mirati Therapeutics [55]. Among the different known mutations, the *KRAS* G12C (glycine 12 to cysteine) mutation has been considered potentially druggable [6]. Several novel covalent direct inhibitors targeting *KRAS* G12C with similar covalent binding mechanisms are now in clinical trials. Both AMG 510 from Amgen and MRTX849 from Mirati Therapeutics covalently bind to *KRAS* G12C at the cysteine at residue 12, keeping *KRAS* G12C in its inactive GDP-bound state and inhibiting *KRAS*-dependent signaling. Both inhibitors are being studied as single agents or in combination with immunotherapy phase 2 trials. In addition, two novel *KRAS* G12C inhibitors, JNJ-74699157 and LY3499446, are being tested [56]. Oral treatment with these compounds shows anti-tumor effects in preclinical models and has shown an ability to maintain *KRAS* in the inactive GDP-bound state. The first in-human study of AMG 510 showed a partial response in patients with non-small cell lung cancer (32.2%) and CRC (7.1%) [57]. In the MRTX849 study, a phase II clinical trial was conducted in 2022 on 64 patients (pancreatic and biliary cancers) with *KRAS* G12C-mutated solid tumors and followed up for 16.8 months (median). Objective responses were observed in 35.1% patients (all partial responses) [58].

Current treatments against *KRAS*-mutated CRC have focused on two paths—the direct and indirect targeting of *KRAS*. Earlier it was seen that the direct targeting of the *KRAS* protein was not practical due to the lack of pocketable hydrophobic spots. The small molecule deltarasin was found to target and bind to the prenyl-binding protein (PDEδ) [59]. This protein inhibited the proliferation of *KRAS*-mutant tumor cells. Utilizing deltarasin as a base, the small molecule compound deltazinone 1 was designed. This compound exhibited higher selectivity and lower unspecific cytotoxicity, displaying strong potential for targeting *KRAS*-mutated tumor cells. This compound showed to inhibit cell proliferation [59].

In a recent study, investigators showed that divarasib (GDC-6036) is an orally bioavailable, covalent *KRAS* G12C inhibitor that turns off its oncogenic signaling by irreversibly locking the protein in an inactive state [60]. In vitro studies have also shown that divarasib is 5 to 20 times as potent and up to 50 times as selective compared to the *KRAS* G12C inhibitors sotorasib and adagrasib (Purkey, H. Discovery of GDC-6036, a clinical stage treatment for *KRAS* G12C-positive cancers. in AACR Annual Meeting, New Orleans, 2022). Single-agent divarasib treatment at 400 mg achieved a confirmed objective response rate (ORR) of 56.4% in patients with non-small cell lung cancer and 35.9% in patients with CRC, with a median progression-free survival (PFS) of 13.1 and 6.9 months, respectively [61]. Divarasib in combination with cetuximab demonstrated a manageable safety profile and promising clinical activity and may represent an effective strategy for overcoming resistance to *KRAS* G12C inhibitors [60].

Lack of clinical follow-up data is a major weakness of our current study. So we cannot comment on the prognostic implications. A comparatively smaller sample size did not allow us to test the associations by further subgrouping by gender or location. With all these limitations in mind, we tried to analyze the data in a way that may help with answering some research questions on pathogenesis and the potential use of precision medicine. Some of the strengths are paired tumor–normal samples from the same individual and the tissues were preserved in an ideal way to stabilize and preserve the RNA and DNA for genome-wide testing. To our knowledge, this is the first study involving patients from a South Asian country to extensively look for the association between *KRAS* mutation, DNA promoter methylation, and transcriptome-wide gene expression in CRC.

## 4. Materials and Methods

The patients and their clinical samples used in this study were included in our previous papers [10,11,15,62]. Between December 2009 and May 2016, tissue samples were collected from 125 CRC patients (male = 72 and female = 53) at different stages (stage I: 25, stage II: 33, and stage III: 67) from the Department of Pathology, Bangabandhu Sheikh Mujib Medical University (BSMMU), Dhaka, Bangladesh. Specimens were collected from resected tumors (CRC tissue) and the surrounding normal-appearing colon tissue (normal tissue) 5–10 cm from the tumor margin for each patient at the time of surgery. None of the patients received any radiotherapy or chemotherapy before surgery. Part of the tissue was kept at −86 °C immediately, and the rest was kept in RNAlater (RNA-stabilizing buffer, Ambion Inc., Austin, TX, USA) and stored at −86 °C. The histopathological diagnosis was conducted independently by two histopathologists at Bangabandhu Sheikh Mujib Medical University (BSMMU), Dhaka, Bangladesh. For each patient, we also abstracted key demographic and clinical data and tumor characteristics from hospital medical records. The tissue samples were sent on dry ice to the molecular genomics lab at the University of Chicago for subsequent analysis. Written informed consent was obtained from all the participants.

The research protocol was approved by the ethical review committee of BSMMU, Dhaka, Bangladesh (BSMMU/2010/10096), and by the IRB of the University of Chicago (10-264-E), Chicago, IL, USA.

### 4.1. DNA Extraction and Quality Control

DNA was extracted from the fresh frozen tissue using a Puregene Core kit (Qiagen, Germantown, MD, USA). An electropherogram from the Agilent BioAnalyzer with Agilent DNA 12,000 chips showed the fragment size to be >10,000 bp. RNA was extracted from RNA Later preserved colonic tissue using Ribopure tissue kit (Ambion, Austin, TX, USA, Cat# AM1924).

### 4.2. Gene Expression

A gene expression assay was conducted on the first 71 pairs (male = 43 and female = 28) of the tissue RNA from the 125 patients. The Genome-Wide Gene Expression Assay we used was the Illumina^®^ TotalPrep RNA Amplification Kit (Ambion, a part of Life Technologies Corporation, Carlsbad, CA, USA) for cRNA synthesis. For the transcriptome-wide gene expression assay, we used the Illumina HT12 v4 BeadChip (Illumina Inc., San Diego, CA, USA) that contains a total of 47,231 probes that cover 31,335 genes. Most of the gene probes target a single isoform (Type-S, n = 28810), some probes target one specific isoform (Type-I, n = 6613), some probes target all known isoforms (Type-A, n = 9563), and some probes target multiple, but not all, known isoforms (Type-M, n = 2245). A total of 71 paired samples (tumor and adjacent healthy tissue) from CRC patients (m = 43, f = 28) were used. Paired tumor and normal samples were processed on the same chip.

### 4.3. DNA Methylation

The DNA methylation study was conducted on all 125 paired tumor samples and corresponding healthy tissue. For this assay, we used a bisulfite conversion kit, EZ-96 DNA Methylation Kit (Zymo Research, Irvine, CA, USA). The HumanMethylation450 DNA analysis BeadChip v1.0 Assay kit was used (Illumina, San Diego, CA, USA). This chip presented 485,577 loci of which 150,254 were in CpG Island, 112,067 were in Shore (0–2 kb from island), 47,114 were in shelf (2–4 kb from the island), and 176,112 were in deep sea (>4 kb from CpG island). Paired samples (CRC and corresponding normal tissue samples) were processed on the same chip to avoid the batch effect. A Tecan Evo robot was used for automated sample processing and the chips were scanned on a single iScan reader. If the intensity of the methylated loci is X and the intensity of the unmethylated loci is Y, then the methylation score (beta value) is X/X + Y. If all are unmethylated (X = 0), then the methylation level is 0/0 + Y = 0. If all the loci are methylated (Y = 0), then the beta value is X/X + 0 = 1. If 50% of the probes are hybridized at the methylated loci and 50% are hybridized at the unmethylated loci, then the methylation score is 50/50 + 50 = 0.5.

### 4.4. Mutation Detection

For the detection of KRAS mutation, a real-time qPCR with HRM analysis was used as described in earlier papers [10,20,63]. The sensitivity and the specificity of the test are 0.97 and 0.87, respectively [63]. We tested 165 paired tumor and adjacent healthy colonic tissue DNA for *KRAS* mutation by HRM analysis. We used the following primers set used by Jesus Gonzalez-Bosquet et al. [63]: *KRAS* F gtg aca tgt tct aat ata gtc aca ttt tc and *KRAS* R ggt cct gca cca gta ata tg. The derived sequence contained mutations in both codon 12 and 13. So a sample having any of the mutations in codon 12 or 13 was identified as *KRAS*-mutant. PCR was carried out on a BioRad (Hercules, CA, USA) CFX 96 thermocycler.

### 4.5. Statistical Analysis

For the statistical analyses, we used the Partek Genomics Suite (version 7.0) (https://www.partek.com/partek-genomics-suite/, accessed on 22 June 2024). We used ANOVA, and Gene set ANOVA as described in previous papers [10,62]. In the GO enrichment analysis, we tested whether the differentially expressed genes (as per the set criteria) fell into a Gene Ontology category more often than expected by chance. We used a chi-square test for comparison. The negative log of the *p*-value for this test was used as the enrichment score. In addition to the GO enrichment analysis, for further statistical comparison of the magnitudes of the differential expression of the “Gene set” in the absence or presence of a factor (mutation), we used “Gene set ANOVA”, which offers an introduction of interaction terms in the model. Gene set ANOVA is a mixed model ANOVA to test the expression of a set of genes (sharing the same category or functional group) instead of an individual gene in different groups. The analysis is performed at the gene level, but the result is expressed at the level of the Gene set category by averaging the member genes’ results. The equation for the model was:Model: Y = μ + T + G + TxG + TxMut + ε
where Y represents the expression status of a Gene set category, μ is the common effect or average expression of the Gene set category, T is the tissue-to-tissue (tumor/normal) effect, G is the gene-to-gene effect, TxG is the differential pattern of gene expression in different tissue types, TxMut is the interaction term, and ε represents the random error.

## 5. Conclusions

We provide evidence of associations between the *KRAS* somatic mutation, transcriptome-wide differential gene expression, and the methylation of gene pathways in CRC. We found that the magnitude of dysregulation of many major gene pathways in CRC were significantly higher in patients with the *KRAS* mutation; and the up- and down-regulation of those dysregulated gene pathways could be correlated with corresponding hypo- and hyper-methylation. In addition, we found that (a) the more pronounced up-regulation of *CDKN2A* in tumors with the *KRAS* mutation may suggest possibility of a better response to anti-*CDKN2A* therapy with Villosol in *KRAS*-mutant CRC; and (b) the more pronounced up-regulation of genes in the proteasome pathway in the CRC tissue especially with the *KRAS* mutation and MSI may suggest a potential role of a proteasome inhibitor in selected CRC patients.

## Figures and Tables

**Figure 1 ijms-25-08094-f001:**
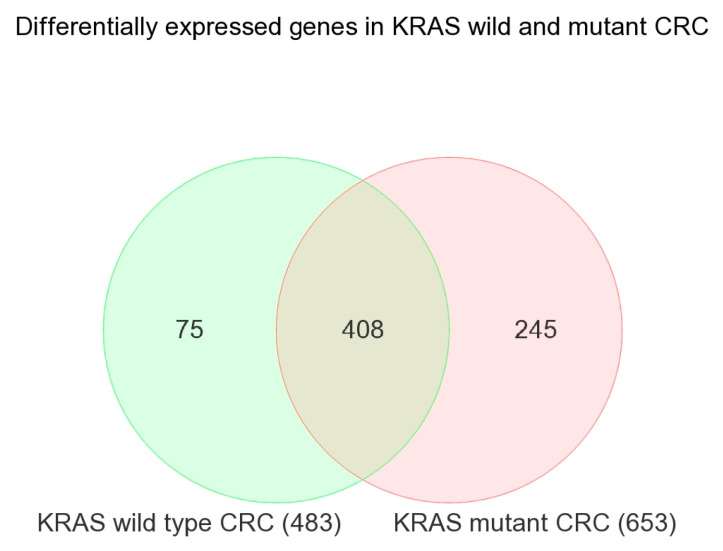
Venn diagram showing the overlap of differentially expressed genes in *KRAS* wild-type CRC (in green) and *KRAS*-mutant CRC (in pink).

**Figure 2 ijms-25-08094-f002:**
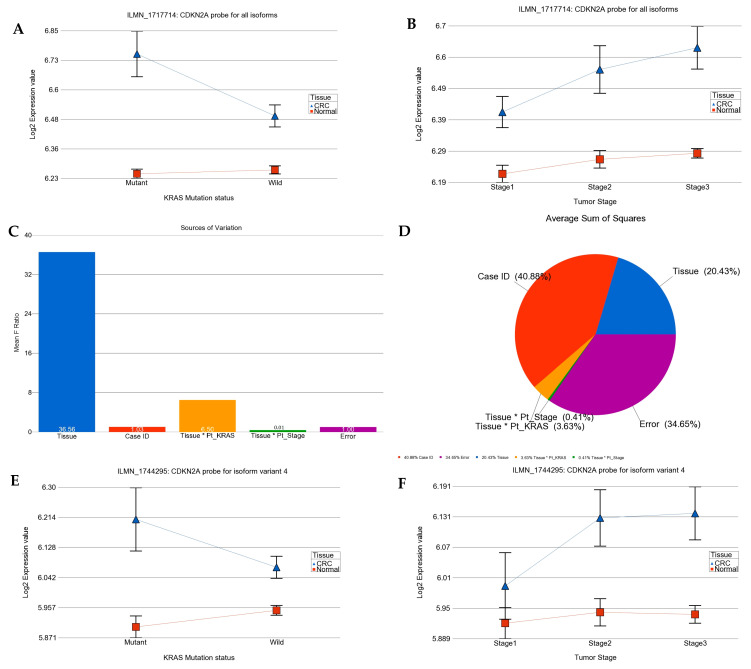
Differential gene expression of *CDKN2A (p16)* in CRC tissue compared to normal tissue by *KRAS* mutation status and by tumor stage. Tumor tissue is shown in blue and normal tissue in red. The top row presents the expression of probe targeting all known isoforms of *CDKN2A* (probe ILMN_1717714). Differential expression by *KRAS* mutation status and by tumor stage are shown on left side (**A**) and right side (**B**), respectively. The source of variation of *CDKN2A* (probe ILMN_1717714) is shown in the second row. The mean F-ratio (F-statistics for the factor/F-statistics for the model error) representing the significance of the factor in the ANOVA model is shown in the bar graph on the left (**C**). The sums of squares in the ANOVA model representing the proportion of the variation explained by the factors are shown as pie chart on the right (**D**). Similarly the expression data for *CDKN2A* probe ILMN_1744295 (probe type I, targeting longest isoform variant 4) by KRAS mutation status and by stage are shown in the third row (**E**,**F**); while the source of variation for this probe ILMN_1744295 is shown in fourth row (**G**,**H**). The expression data for *CDKN2A* probe ILMN_1757225 (probe type I, targeting only isoform variant 3) by *KRAS* mutation status and by stage are shown in the fifth row (**I**,**J**); while the source of variation for the probe ILMN_1757225 is shown in sixth row (**K**,**L**).

**Figure 3 ijms-25-08094-f003:**
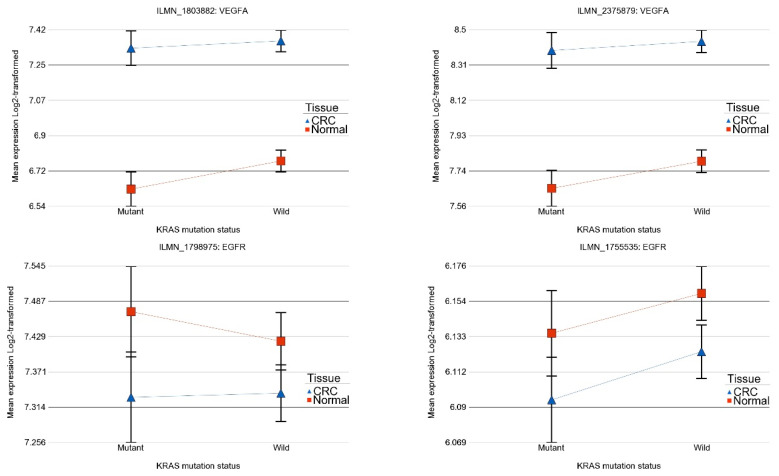
Differential gene expression among *KRAS*-mutant and wild-type in CRC and normal tissue in different forms of interaction. The top row shows two gene probes of vascular endothelial growth factor A (*VEGFA*) gene where the gene is up-regulated in CRC than in normal tissue, irrespective of *KRAS* mutational status. The magnitude of up-regulation was not significantly different among patients with or without KRAS mutation. The bottom row shows two gene probes of epidermal growth factor receptor (*EGFR*) gene where there is no significant difference between CRC and normal tissue in both *KRAS* -mutant and wild-type.

**Figure 4 ijms-25-08094-f004:**
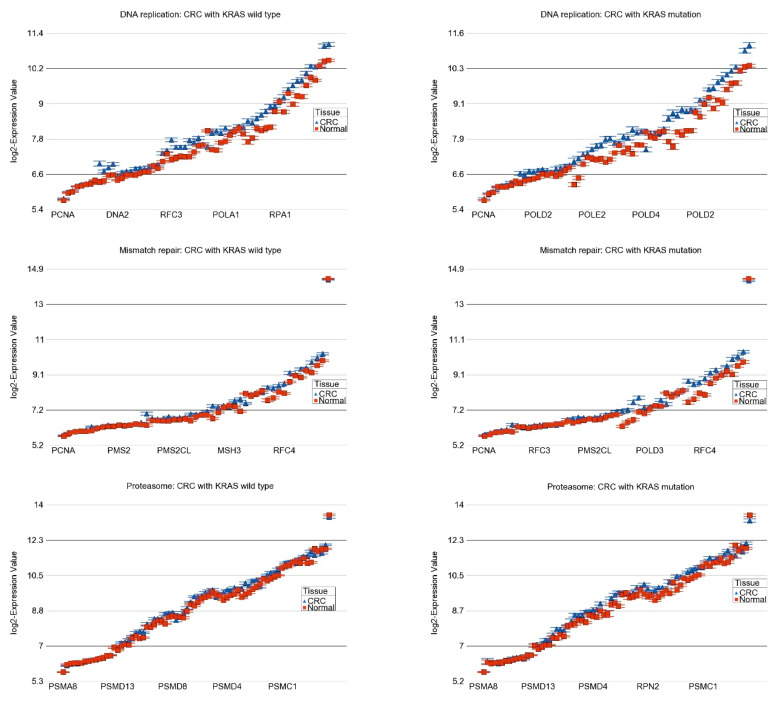
Differential expression of pathways in CRC (in blue) compared to normal tissue (in red) by *KRAS* mutational status. CRC with *KRAS* wild-type is shown in the left panel and CRC with *KRAS* mutation is shown in the right panel. The up-regulation of genes in DNA replication pathway, mismatch repair pathway, and proteasome pathway in CRC were more pronounced in the presence of *KRAS* mutation (ANOVA interaction *p*-values are shown in Table 2). The down-regulation of genes involved in intestinal immune network for IgA production was more pronounced in *KRAS* mutant CRC. Genes are arranged on the X-axis by expression level, and the log_2_-Expression Value is shown on the Y-axis. Gene symbols for all the genes could not be shown on the X-axis.

**Figure 5 ijms-25-08094-f005:**
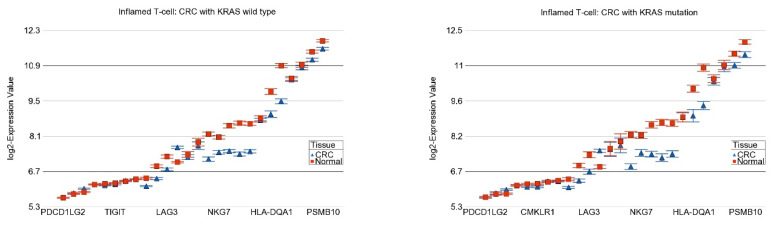
Differential expression of the inflamed T cell genes in RC (in blue) compared to normal tissue (in red) by *KRAS* mutational status. CRC with *KRAS* wild-type is shown in the left panel and CRC with *KRAS* mutation is shown in the right panel. The down-regulation in CRC was more pronounced in the presence of *KRAS* mutation (ANOVA interaction *p* = 0.003). Genes are arranged on the X-axis by expression level, and the log_2_-Expression Value is shown on the Y-axis. Gene symbols for all the genes could not be shown on the X-axis.

**Table 1 ijms-25-08094-t001:** Differential expression of gene pathways in CRC vs. normal tissue—combined analysis. The result is arranged by the FC in CRC tissue. The highlighted pathways are the pathways with significant interaction with *KRAS* mutation status (as shown in the next step in Table 2).

GO Description	Fold Change	(95% CI)	*p*-Value
DNA replication	1.23	(1.22–1.25)	0.00 × 10^0^
Ribosome biogenesis in eukaryotes	1.18	(1.17–1.19)	0.00 × 10^0^
RNA polymerase	1.16	(1.15–1.17)	4.82 × 10^−178^
Mismatch repair	1.16	(1.14–1.17)	8.37 × 10^−165^
Bladder cancer	1.14	(1.12–1.16)	4.01 × 10^−61^
ECM-receptor interaction	1.14	(1.13–1.15)	4.91 × 10^−147^
One carbon pool by folate	1.13	(1.12–1.15)	1.08 × 10^−51^
Nucleotide excision repair	1.13	(1.12–1.14)	1.84 × 10^−187^
Vitamin B6 metabolism	1.12	(1.08–1.16)	4.13 × 10^−10^
Proteasome	1.11	(1.11–1.12)	1.56 × 10^−128^
Malaria	1.11	(1.09–1.13)	9.13 × 10^−28^
Propanoate metabolism	−1.11	(−1.12–−1.09)	5.43 × 10^−60^
Gastric acid secretion	−1.11	(−1.12–−1.1)	2.11 × 10^−117^
Antigen processing and presentation	−1.11	(−1.12–−1.1)	3.98 × 10^−92^
Collecting duct acid secretion	−1.11	(−1.13–−1.09)	5.59 × 10^−31^
Type I diabetes mellitus	−1.11	(−1.14–−1.09)	4.77 × 10^−27^
Fat digestion and absorption	−1.11	(−1.13–−1.1)	2.35 × 10^−40^
Galactose metabolism	−1.12	(−1.13–−1.1)	3.49 × 10^−52^
Butanoate metabolism	−1.12	(−1.14–−1.1)	9.13 × 10^−48^
Ether lipid metabolism	−1.12	(−1.14–−1.11)	3.41 × 10^−69^
Citrate cycle (TCA cycle)	−1.12	(−1.14–−1.11)	6.91 × 10^−79^
Other glycan degradation	−1.12	(−1.14–−1.1)	2.98 × 10^−31^
Bile secretion	−1.13	(−1.14–−1.12)	3.50 × 10^−91^
Valine, leucine, and isoleucine degradation	−1.13	(−1.15–−1.12)	1.72 × 10^−82^
Porphyrin and chlorophyll metabolism	−1.14	(−1.16–−1.12)	3.21 × 10^−55^
Autoimmune thyroid disease	−1.14	(−1.16–−1.13)	3.50 × 10^−74^
Metabolism of xenobiotics by cytochrome P450	−1.15	(−1.16–−1.13)	7.19 × 10^−86^
Primary immunodeficiency	−1.16	(−1.17–−1.14)	1.20 × 10^−137^
Chemical carcinogenesis	−1.16	(−1.17–−1.15)	4.30 × 10^−135^
Mineral absorption	−1.17	(−1.18–−1.15)	3.03 × 10^−117^
Steroid hormone biosynthesis	−1.17	(−1.18–−1.15)	7.99 × 10^−91^
Graft-versus-host disease	−1.17	(−1.2–−1.15)	5.87 × 10^−60^
Fatty acid degradation	−1.18	(−1.19–−1.16)	3.96 × 10^−149^
Drug metabolism-cytochrome P450	−1.19	(−1.21–−1.17)	1.07 × 10^−116^
Aldosterone-regulated sodium reabsorption	−1.21	(−1.23–−1.2)	1.55 × 10^−172^
Retinol metabolism	−1.22	(−1.24–−1.2)	6.74 × 10^−139^
Ascorbate and aldarate metabolism	−1.22	(−1.25–−1.2)	2.50 × 10^−61^
Synthesis and degradation of ketone bodies	−1.23	(−1.27–−1.19)	2.16 × 10^−30^
Asthma	−1.23	(−1.26–−1.2)	2.52 × 10^−55^
Allograft rejection	−1.23	(−1.26–−1.21)	3.88 × 10^−82^
Intestinal immune network for IgA production	−1.25	(−1.27–−1.22)	8.48 × 10^−134^
Caffeine metabolism	−1.26	(−1.31–−1.22)	2.71 × 10^−34^
Pentose and glucoronate interconversions	−1.27	(−1.29–−1.24)	3.03 × 10^−93^
Sulfur metabolism	−1.29	(−1.33–−1.25)	1.75 × 10^−52^
Proximal tubule bicarbonate reclamation	−1.34	(−1.37–−1.31)	1.67 × 10^−120^
Nitrogen metabolism	−1.72	(−1.81–−1.63)	2.45 × 10^−89^

**Table 2 ijms-25-08094-t002:** Differential expression of pathways in CRC vs. normal tissue by *KRAS* mutation status. The result is arranged by the FC in KRAS-mutant CRC.

GO Description	*KRAS* Wild-Type		*KRAS*-Mutant		Interaction-*p*
	Fold Change	(95% CI)	Fold Change	(95% CI)	
DNA replication	1.21	(1.19–1.22)	1.27	(1.25–1.3)	1.86 × 10^−05^
Ribosome biogenesis in eukaryotes	1.16	(1.15–1.17)	1.21	(1.19–1.23)	4.35 × 10^−07^
Mismatch repair	1.13	(1.12–1.15)	1.19	(1.17–1.22)	5.39 × 10^−06^
RNA polymerase	1.15	(1.13–1.16)	1.19	(1.17–1.21)	4.25 × 10^−03^
One carbon pool by folate	1.11	(1.09–1.13)	1.18	(1.15–1.22)	3.10 × 10^−03^
Nucleotide excision repair	1.12	(1.11–1.13)	1.15	(1.14–1.17)	5.67 × 10^−04^
Proteasome	1.10	(1.09–1.11)	1.14	(1.12–1.16)	9.07 × 10^−11^
Aminoacyl-tRNA biosynthesis	1.09	(1.07–1.1)	1.14	(1.12–1.16)	7.85 × 10^−06^
Base excision repair	1.09	(1.07–1.1)	1.13	(1.11–1.15)	2.49 × 10^−04^
ECM-receptor interaction	1.14	(1.13–1.15)	1.12	(1.1–1.14)	8.17 × 10^−05^
Steroid biosynthesis	1.06	(1.03–1.09)	1.11	(1.07–1.15)	1.31 × 10^−04^
Homologous recombination	1.07	(1.06–1.08)	1.10	(1.08–1.12)	5.15 × 10^−03^
Inflammatory bowel disease (IBD)	−1.07	(−1.09–−1.06)	−1.10	(−1.13–−1.08)	1.55 × 10^−02^
Staphylococcus aureus infection	−1.09	(−1.11–−1.08)	−1.11	(−1.13–−1.08)	3.95 × 10^−03^
Antigen processing and presentation	−1.11	(−1.12–−1.1)	−1.12	(−1.14–−1.09)	2.81 × 10^−05^
Renin secretion	−1.09	(−1.1–−1.08)	−1.12	(−1.14–−1.1)	7.59 × 10^−03^
Viral myocarditis	−1.09	(−1.11–−1.08)	−1.12	(−1.15–−1.1)	9.38 × 10^−04^
Tyrosine metabolism	−1.08	(−1.1–−1.06)	−1.12	(−1.16–−1.09)	9.63 × 10^−04^
Type I diabetes mellitus	−1.11	(−1.13–−1.08)	−1.13	(−1.18–−1.09)	2.41 × 10^−03^
Autoimmune thyroid disease	−1.13	(−1.15–−1.11)	−1.17	(−1.2–−1.14)	4.30 × 10^−04^
Primary immunodeficiency	−1.15	(−1.16–−1.13)	−1.19	(−1.21–−1.16)	1.19 × 10^−02^
Graft-versus-host disease	−1.16	(−1.19–−1.14)	−1.20	(−1.25–−1.16)	8.67 × 10^−04^
Mineral absorption	−1.15	(−1.17–−1.13)	−1.20	(−1.23–−1.18)	5.52 × 10^−03^
Allograft rejection	−1.21	(−1.24–−1.18)	−1.28	(−1.33–−1.23)	1.60 × 10^−04^
Asthma	−1.21	(−1.25–−1.17)	−1.29	(−1.36–−1.23)	9.46 × 10^−03^
Intestinal immune network for IgA production	−1.23	(−1.26–−1.21)	−1.29	(−1.33–−1.25)	7.53 × 10^−03^

**Table 3 ijms-25-08094-t003:** Interaction of *KRAS* mutation and MSI status for differential gene expression of several pathways in CRC compared to corresponding normal tissue.

			DNA Replication Pathway		Proteasome Pathway		Mismatch Repair Pathway	
			*KRAS* Mut − ve	*KRAS* Mut + ve	*KRAS* Mut − ve	*KRAS* Mut + ve	*KRAS* Mut − ve	*KRAS* Mut + ve
MSI Status	MSI	FC	1.19	1.27	1.06	1.18	1.12	1.20
		95% CI	(1.17–1.22)	(1.22–1.32)	(1.04–1.08)	(1.14–1.22)	(1.09–1.14)	(1.15–1.25)
	MSS	FC	1.22	1.27	1.12	1.12	1.14	1.19
		95% CI	(1.2–1.23)	(1.24–1.3)	(1.1–1.13)	(1.1–1.15)	(1.13–1.16)	(1.16–1.22)
Interaction-*p*			7.51 × 10^−16^		4.71 × 10^−29^		2.17 × 10^−10^	

**Table 4 ijms-25-08094-t004:** Methylation status of up-regulated gene pathways with interaction with *KRAS*.

GO Description	*KRAS* Wild-Type		*KRAS*-Mutant		Interaction-*p*
	*p*-Value	Delta Beta	*p*-Value	Delta Beta	
Steroid biosynthesis	3.12 × 10^−11^	−0.0149	2.43 × 10^−10^	−0.0232	5.72 × 10^−02^
RNA polymerase	4.19 × 10^−16^	−0.0090	1.34 × 10^−13^	−0.0134	8.67 × 10^−04^
One carbon pool by folate	6.67 × 10^−05^	−0.0056	7.73 × 10^−06^	−0.0104	1.75 × 10^−04^
Aminoacyl-tRNA biosynthesis	2.00 × 10^−49^	−0.0084	6.50 × 10^−20^	−0.0085	1.48 × 10^−09^
Proteasome	2.33 × 10^−36^	−0.0082	1.48 × 10^−11^	−0.0072	1.58 × 10^−12^
Base excision repair	7.55 × 10^−13^	−0.0071	1.34 × 10^−05^	−0.0070	2.99 × 10^−11^
Mismatch repair	9.67 × 10^−06^	−0.0054	1.81 × 10^−02^	−0.0047	3.11 × 10^−08^
DNA replication	1.28 × 10^−16^	−0.0067	9.83 × 10^−04^	−0.0044	6.96 × 10^−15^
Homologous recombination	3.71 × 10^−25^	−0.0094	3.76 × 10^−03^	−0.0043	2.62 × 10^−18^
Nucleotide excision repair	1.25 × 10^−04^	−0.0031	6.99 × 10^−01^	−0.0005	2.72 × 10^−18^
Ribosome biogenesis in eukaryotes	2.46 × 10^−02^	−0.0016	2.88 × 10^−04^	0.0043	2.92 × 10^−27^
ECM-receptor interaction	6.05 × 10^−23^	0.0201	9.36 × 10^−19^	0.0295	7.49 × 10^−10^

**Table 5 ijms-25-08094-t005:** Methylation status of down-regulated gene pathways with interactions with *KRAS*.

GO Description	*KRAS* Wild-Type		*KRAS*-Mutant		Interaction *p*
	*p*-Value	Delta Beta	*p*-Value	Delta Beta	
Asthma	8.16 × 10^−26^	0.072	2.73 × 10^−16^	0.090	2.88 × 10^−02^
Intestinal immune network for IgA production	3.34 × 10^−30^	0.030	7.54 × 10^−39^	0.057	8.17 × 10^−23^
Type I diabetes mellitus	5.50 × 10^−194^	0.045	6.78 × 10^−106^	0.054	3.50 × 10^−09^
Allograft rejection	5.50 × 10^−194^	0.045	6.78 × 10^−106^	0.054	3.50 × 10^−09^
Graft-versus-host disease	5.50 × 10^−194^	0.045	6.78 × 10^−106^	0.054	3.50 × 10^−09^
Autoimmune thyroid disease	5.88 × 10^−171^	0.043	3.43 × 10^−92^	0.051	7.91 × 10^−08^
Antigen processing and presentation	1.01 × 10^−187^	0.031	1.14 × 10^−98^	0.036	5.64 × 10^−11^
Primary immunodeficiency	7.71 × 10^−16^	0.015	3.21 × 10^−24^	0.030	5.23 × 10^−18^
Inflammatory bowel disease (IBD)	1.34 × 10^−33^	0.019	8.04 × 10^−31^	0.029	1.56 × 10^−11^
Viral myocarditis	1.03 × 10^−93^	0.017	4.71 × 10^−53^	0.021	3.51 × 10^−15^
Renin secretion	5.43 × 10^−46^	0.014	1.56 × 10^−10^	0.010	8.34 × 10^−04^
Tyrosine metabolism	4.23 × 10^−01^	0.001	4.78 × 10^−01^	0.002	6.32 × 10^−09^
Mineral absorption	1.12 × 10^−05^	−0.005	6.48 × 10^−01^	−0.001	2.28 × 10^−07^
Staphylococcus aureus infection	9.65 × 10^−07^	−0.017	1.48 × 10^−02^	−0.014	2.10 × 10^−02^

## Data Availability

All the supporting data are presented in the tables presented in the main manuscript and Appendix A.

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
