# Peer review of "Association of KRAS Mutation and Gene Pathways in Colorectal Carcinoma: A Transcriptome- and Methylome-Wide Study and Potential Implications for Therapy"

_ijms, 2024, doi:10.3390/ijms25158094_

Round 1

Reviewer 1 Report

Comments and Suggestions for Authors

The authors present an interesting manuscript on the Association of KRAS mutation and gene pathways in Colorectal carcinoma. The manuscript is well presented. Objectives are clearly presented. Introduction is correct. Material and methods are correct. Results are clearly presented and easy to follow. 

There are some questions:

Abstract: There is no mention to Results and Conclusions.

Material and Methods: Authors do not mention if the study was examined and approved by Local Ethics Committee

Discussion Section does not exist

Conclusions: There are no clear and objectively expressed conclusions

Please find the additional comments in the attached file.

Reviewer 2 Report

Comments and Suggestions for Authors

The original paper's authors used KRAS wild-type and KRAS mutant colorectal cancer samples from an Asian population to study gene expression and methylation in the molecular pathways they were looking into.
They were interested in how the KRAS mutational status influences the expression pattern of each affected pathway and whether these changes are associated with promoter methylation abnormalities. The goal was to name a targeted treatment for CRC.
The number of samples tested is adequate, and the study design is appropriate. The methods used, as well as the technical description, are appropriate.
Tables and figures help to understand the subject and the results.
Their results show that KRAS status influences the expression of the functional gene clusters under study and is also associated with their methylation status in CRC. They suggest the identification of molecular targets (e.g., CDKN2A-TP53-PI3K/Akt axis) that could form the basis for future targeted therapy.
The results speak for themselves, and I can find nothing to object to in the paper. 
However, I highlight as a fundamental problem the fact that the authors did not perform validation studies on an independent sample. Moreover, I observed that the authors failed to test the targeted inhibition of the CDKN2A-TP53-PI3K/Akt axis in KRAS mutant and WT cell lines. I believe that without them, the article in its current form does not add much to the basic research results of molecular therapy for CRC. 

Reviewer 3 Report

Comments and Suggestions for Authors

KRAS is a commonly mutated oncogene in colorectal cancer (CRC). This peer-reviewed study investigated transcriptome and DNA methylation differences between KRAS-positive and KRAS-negative CRC samples, identifying several differentially expressed pathways previously reported in the literature.

The impact of KRAS mutations on CRC has been extensively studied, so the methodology and findings of this study are not particularly novel. For instance, it is well-established that mutated KRAS can lead to the upregulation of p16 in CRC, and not only.

KRAS mutations in CRC typically occur at codons 12 or 13, each mutation affecting tumor cell metabolism differently. However, this study lacks specific information on the KRAS mutation status, which should be included as a factor in the analysis.

The mutational status of other genes, such as TP53, should be considered too.

While the study collected samples from different tumour stages, it is unclear which samples were used for RNA sequencing and bisulfite methylation analysis. The tumour stage is an important factor in transcriptome studies. Information on the level of KRAS expression in the KRAS-positive group is also missing. Likely, the analysed samples showed different KRAS expression.

Two major KRAS isoforms 4A and 4B were identified. This study focuses on the KRAS4A isoform, I assume. It would be nice to see if the expression of KRAS4B corresponds to KRAS4A.

The study is primarily descriptive and lacks follow-up or functional analyses. Comparative organoid studies on samples with different KRAS mutations would greatly enhance the findings.

Round 2

Reviewer 1 Report

Comments and Suggestions for Authors

The new version of the manuscript is sufficiently improved to warrant publication in IJMS 

Reviewer 2 Report

Comments and Suggestions for Authors

Thank you for the correct explanation and answer from the authors.  The revised version of the manuscript is now suitable for publication. 

Reviewer 3 Report

Comments and Suggestions for Authors

The manuscript has been partially revised, and it reads now better.

I recommend acceptance for publication.